# A framework to measure the taxonomic of economic anchor: A case study of the Three Seas Initiative countries

**Magdalena Kozera-Kowalska**[1]*, **Jarosław Uglis**[2], **Jarosław Lira**[1]

**1** Faculty of Economics, Poznań University of Life Sciences, Poznań, Poland, **2** Faculty of Veterinary Medicine and Animal Science, Poznań University of Life Sciences, Poznań, Poland

☯ These authors contributed equally to this work.
* magdalena.kozera@up.poznan.pl

## Abstract

This research is aimed at determining the characteristics of the current level of entrepreneurial potential of the Three Seas Initiative (3SI) countries, the ability to overcome the consequences of extraordinary events, such as COVID-19 and prospects for the return to an accelerated development once the destabiliser of the economic system, the coronavirus pandemic, has ceased. Eurostat, World Bank and the World Economic Forum data for 2015–2019 were used for the purpose of the research. The research was divided into three stages, i.e. assessment of economic development on the basis of a synthetic ratio of economic anchor development, for which a relative benchmark method based on spatial median (so-called L1 median or Weber point) was used, identification of conditions for the development of entrepreneurial capacity and statistical analysis showing the correlation between economic anchor measures and selected factors of the 3SI countries economic development. Our study found that the entrepreneurial capacity of the 3SI countries in 2015–2019 was determined by nine characteristics, belonging to six areas, i.e. local economy, demographic situation, social situation, trade exchange, innovation and tourism economy. The entrepreneurial potential of the 3SI countries was spatially diversified, and its development was determined, among others, by the entrepreneurial activity of residents (entrepreneurship index) and the conditions for running a business.

## Introduction

The coronavirus pandemic and its socio-economic consequences have triggered a broad discussion on the functioning of world economies both during and after the pandemic. COVID-19 has become a test of the sustainability of economic systems, the efficiency of business environment institutions, and the effectiveness of the impact of the entire spectrum of factors determining the level and pace of development [1, 2]. It is particularly relevant to the relatively new economic arrangements in almost all regions of the world [3–5]. One such group is the Three Seas Initiative (3SI), which brings together 12 Central European countries —Austria, Bulgaria, Croatia, Czechia, Estonia, Hungary, Latvia, Lithuania, Poland, Romania,

**Funding:** The authors received no specific funding for this work.

**Competing interests:** The authors have declared that no competing interests exist.

Slovakia and Slovenia. These countries have different economic potential and natural conditions and differently formulated development goals [6–9].

Understanding the method of assessing entrepreneurial potential is a utilitarian challenge and applies not only to the Three Seas countries, but also to other, especially less affluent ones. The gravity of the problem is significant because, as researchers indicate, the developmental disproportions of the countries forming the 3SI are very deep and the resulting problems are so important that the territory of the Three Seas Initiative can be treated as a "specific area of structural negligence" [10–13]. Diagnosis of the problem is also based on research justification due to the fact that there are few individual studies on the spatial distribution of entrepreneurial potential factors that can have a significant impact on restoring the efficiency of the economy deregulated by exceptional factors outside the economic system [14].

The research undertaken is aimed at determining the characteristics of the current level of entrepreneurial potential of the 3SI countries, ability to overcome the consequences of extraordinary events, including those caused by COVID-19, prospects for the return to an accelerated development once the destabiliser of the economic system (coronavirus pandemic), has ceased.

In order to achieve this goal, a proprietary measure of entrepreneurial potential was constructed, calling it an economic anchor [15], which was determined by one of the methods of relative taxonomy based on the construction of a relative synthetic measure based on spatial median.

By creating it, the aim was to answer the following research questions:
What factors shaping entrepreneurial potential are particularly important?
How spatially diverse is the entrepreneurial potential in the countries?
What factors are conducive to developing this potential?

These issues are presented in subsequent parts of the work, ranging from systematizing the concepts and measures of entrepreneurial potential and factors shaping it (Background of the research), through discussing the conditions for the creation of the Three Seas Initiative (3SI) and the capacities of its members (The Three Seas Initiative as a research area), presentation of research methods (Material and method) and presentation of research results (Research results) and discussion of them (Discussion) and summary of the results achieved (Conclusions).

## Background of the research

The complexity of the research problem makes obtaining information and selecting appropriate research methods an innovative task. This is due to the fact that there are few scientific studies on the factors shaping the entrepreneurial potential of regions, including the relatively new 3SI grouping established in 2015. Therefore, out of necessity, the study was conducted based on the per analogy principle [16, 17], enabling formulation of conclusions and generalizations based on the results of other authors research on regional cooperation, international organizations and groups (especially concerning the Visegrád Group. Moreover, studies related to support instruments and barriers to economic development in EU countries were an interesting source of knowledge.

The entrepreneurial potential of territorial entities (regions, countries, groups) is a total set of natural and formal-legal conditions on four levels, i.e. individual (man), social (group), organizational (company, institution) and overall (economic system) [25]. Their configuration creates more or less favourable conditions for entrepreneurial activity, which in material terms constitutes the basic form of economic activity and its effects are subject to general market laws [18–20]. In a broad sense, entrepreneurship is an organised, social process of creating and using opportunities [21, 22], the effects of which are identified at the individual level

(employee, entrepreneur, income, career), micro-economic level (enterprises, development, products) and macroeconomic level (economy, jobs, taxes) [23]. This means that the improvement of conditions for entrepreneurial activity not only affects the entrepreneurial potential of the country, but also increases its attractiveness and competitiveness on the international market. It is important to distinguish between factors shaping entrepreneurship, which are usually country-specific or universal for all countries, and factors shaping entrepreneurial potential, which are only country-specific [24–26]. For the analysis of entrepreneurial potential, it is also important to refer to the broader context of their location in economic structures, e.g. in the EU. This gives rise to comparisons on the benchmarking basis, i.e. changes taking place within one of the countries to the whole system, or the country taken as a reference point [27, 28]. This makes it possible, inter alia, to determine the competitive position of the country or group of countries concerned in relation to other economic systems.

Therefore, for the needs of the research it has been assumed that the entrepreneurial potential of a given area will be understood as a total set of factors influencing the level of social and economic activity of citizens and legal organisational entities determined by the profile and character of this territory and its ability to actively participate in creating a modern innovative economy based on knowledge [29, 30]. A specially constructed economic anchor is used as a measure of this set, describing the entrepreneurial potential in a given country. It is a synthetic measure created by combining other statistical values, which are measurements of different parameters. It enables the division of different business entities into groups and a uniform, objective measurement and comparison of the analysed categories in a manner consistent with the defined objective. Determination of the economic anchor consisted in the analysis of a multidimensional object, i.e. 3SI, according to the criterion of the adopted research objective, i.e. level of the entrepreneurial potential. It is worth noting that the term anchor is usually used in national policies, in which they mean a framework (mostly financial one) of conducting activities determined by the decisions of public authority of a state [31]. The importance of the problem is due to the fact that in the era of the COVID-19 pandemic and the economic slowdown caused by it, which has the characteristics of an economic crisis, it will be necessary to make non-standard decisions based on unconventional information about unusual circumstances. Lack of knowledge in this area may result both in the scale of individual countries and their groups (e.g. EU), development marginalization with consequences on a global scale [32, 33].

## Materials and methods

### Three Seas Initiative as a research entity

The Three Seas Initiative, (TSI or 3SI) is an initiative that is intended to highlight the potential of economic cooperation between the Adriatic, Baltic and Black Seas. The Group was established on the initiative of Poland and Croatia in 2015, and the objectives of the agreement have become primarily: developing cooperation in areas such as energy and transport, improving digital communication, striving to strengthen levels of security and competitiveness and equalizing the economic development of the signatories. The objectives adopted by 3SI are fully compatible with the three basic EU documents, i.e. 2030 Energy strategy, Digital single market and Roadmap to a Single European Transport Area [7]. The objectives of 3SI are to be achieved by a special Fund and the Three Seas Chamber of Commerce Network.

According to some authors [34], the revitalisation of the Visegrád Group (V4), and in particular the change of its priorities (seeking real influence on EU decisions) [7]. Other authors see in 3SI the implementation of the Polish geopolitical concept of the Intermarium, which dates back to the times before World War II and aims to strengthen Poland's position as a country geographically situated between two antagonistic powers, i.e. Germany and Russia [9,

35]. Regardless of the difference in approach, what is important is that the Three Seas is organizing pre-existing forms of cooperation between Central European countries, (i.e. the Central European Initiative (CEI), Council of the Baltic Sea States, Organisation of Black Sea Economic Cooperation and others), whose real impact has been decreasing over the years. The Initiative thus gives a new dimension to previous cooperation, which for various reasons has ceased to satisfy the parties involved.

It still seems to be a matter of discussion both to estimate the full capacities of the Three Seas, not only in the economic and social aspect, but also in the political one; and to specify its objectives. The vision of this project as an informal infrastructure project, which is to result in a north-south communication link (alternative and/or competitive to the existing east-west line) comes to the fore [36]. The creators of the initiative, including Poland, clearly stated that 3SI is to be a tool to strengthen the EU, by improving cooperation between Central and Eastern European countries.

From the point of view of research on the factors shaping entrepreneurial potential, it is important that the founding declarations include cooperation in the areas of economics, transport, energy infrastructure, environmental protection, research and development and digital communication. The aim is to enable the member states of the Initiative to access, inter alia, the Trans-European Transport Network Executive Agency (TEN-T EA) [37, 38]. This would enable an increase in the transfer of tangible and intangible resources between the 3SI countries, resulting in acceleration of business development.

When analysing the entrepreneurial potential in 2015–2017, it can be seen that 3SI countries have similar sizes of selected indicators, as exemplified by the number of enterprises per 1,000 inhabitants), which do not differ significantly for both the EU-28 and EU-27, and are even higher in 2017 (Table 1).

Similar conclusions are provides the analysis of the average annual growth rate of the number of economic operators in the Three Seas countries compared to the EU. However, the source values indicate significant diversification, which is reflected in the calculated values of the synthetic economic anchor ratio.

## Material and method

The source material for our own research was secondary data coming mainly from Eurostat, World Bank and World Economic Forum databases. In addition, the European Commission

**Table 1. Entrepreneurship in 3SI countries compared to the EU.**

| Specification | Years | | |
|---|---|---|---|
| | **2015** | **2016** | **2017** |
| Number of enterprises per 1,000 inhabitants | | | |
| 3SI | 54 | 56 | 58 |
| EU-27 | 55 | 56 | 57 |
| EU-28 | 54 | 55 | 56 |
| % share of 3SI residents | | | |
| EU-27 | 25.2 | 25.1 | 25.1 |
| EU-28 | 22.0 | 21.9 | 21.8 |
| % share of 3SI companies | | | |
| EU-27 | 24.7 | 24.7 | 25.4 |
| EU-28 | 22.7 | 22.5 | 23.2 |

Source: EUROSTAT.

**Table 2. Selected diagnostic features.**

| Item | Designation | Secondary criterion | Name of the characteristic |
|---|---|---|---|
| 1 | $X_1$ | 1.Local economy | Value added per employee |
| 2 | $X_2$ | | Share of employees in agriculture, forestry and fishing |
| 3 | $X_3$ | | Share of employees in manufacturing |
| 4 | $X_4$ | | Share of employees in trade |
| 5 | $X_5$ | | Share of persons working in accommodation and food service activities |
| 6 | $X_6$ | 2.Demographic situation | Population density |
| 7 | $X_7$ | 3.Social situation | Share of unemployed in the total number of active people |
| 8 | $X_8$ | | Share of persons of working age (15–64) |
| 9 | $X_9$ | 4.Trade exchange | Export of goods and services (% in GDP) |
| 10 | $X_{10}$ | | Imports of goods and services (% of GDP) |
| 11 | $X_{11}$ | 5.Innovation | Innovation index |
| 12 | $X_{12}$ | 6.Tourism economy | Use of accommodation capacity |
| 13 | $X_{13}$ | | Tourist traffic intensity-Schneider's index |
| 14 | $X_{14}$ | | Tourist traffic intensity-Charvat's index |

Source: own research.

communications were used. The collected factual material was from 2015–2019. The study was divided into three stages. The first one consisted in the assessment of the economic development on the basis of a synthetic anchor development measure, for the construction of which 14 diagnostic features grouped according to six detailed criteria were initially analysed (Table 2). Diagnostic variables have been acquired for the EU and Three Seas countries for the years 2015–2019.

A relative benchmark method based on spatial median was used to construct a synthetic measure of economic anchor. Selection of diagnostic features was made on the basis of substantive and statistical premises, assuming those, which are closely related to the subject of the research. The statistical study was based on the analysis of diagonal elements of the inverse R correlation matrix in order to avoid excessive correlation of the assumed diagnostic features. For this purpose, destimulants were transformed into stimulants by means of quotient transformation, and then the values of individual features (If a feature is excessively correlated with the others, the diagonal elements of the inverse $\mathbf{R}^{-1}$matrix significantly exceed the value of 10, which is a symptom of bad numerical conditioning of the $\mathbf{R}$ matrix [39] were relativized, which were used to construct relative matrices in subsequent periods of time. These matrices formed basis for creating average (positional approach) relative matrices in 2015–2019. It should be noted that in relative taxonomy methods there is no features normalisation. Taxonomic values of the synthetic economic anchor measure were used to linearly order countries and to distinguish typological classes bringing together countries from (very) high to (very) low relative development levels. Analysing the differences starting from the difference between the second and first country, etc., the first observed much higher value of such a difference (compared to the previous value) allowed to isolate the relative typological class with the highest level of development. The differences between the successive countries made it possible to determine the next classes.

The values of the synthetic economic anchor measure may be greater or less than 1 and allow the relative position of the country in relation to other countries covered by the research to be determined. The lower the value of the measure is than 1, the greater is the relative advantage of the country compared to the others in terms of synthetic assessment in a period

or moment of time t. In turn, the greater the value of the measure is than 1, the greater is the relative delay, i.e. the weaker is the position of the country in relation to others [more 40, 41].

The median vector used in the research is called spatial median, is also called L1 or Weber point [42]. The following assumption is made when creating it: let $K\_n^m$ = {$X\_1$ ⟦,X⟧ _2,···, $X\_n$,}∈R^m be a set of n vectors for observation of m-featured objects and let Θ ^∈R^m be such a vector that is a solution to the optimization problem

$$T(\hat{\mathbf{\Theta}}, K_n^m) = \min_{\mathbf{\Theta} \in R^m} T(\mathbf{\Theta}, K_n^m)$$ (1)

where: the objective function of this issue takes the form

$$T(\mathbf{\Theta}, K_n^m) = \sum_{i=1}^{n} \left[ \sum_{j=1}^{m} (x_{ij} - \theta_j)^2 \right]^{1/2}$$ (2)

where: $X\_i$ = ($x\_i1, x\_i2, \cdots, x\_im$) ^',i = 1,2,···,n and Θ = ($\theta\_1, \theta\_2, \cdots, \theta\_m$) ^'.

In the relevant study, the data matrices are assumed to be as follows:

$$\mathbf{X}_t = \begin{pmatrix} x_{11t} & x_{12t} & \cdots & x_{1mt} \\ x_{21t} & x_{22t} & \cdots & x_{2mt} \\ \vdots & \vdots & \vdots & \vdots \\ x_{n1t} & x_{n2t} & \cdots & x_{nmt} \end{pmatrix}$$ (3)

where: $x\_ijt$ represents the observation over period t for the i-th diagnostic feature (j = 1,2,..., m) in the i-th spatial entity (i = 1,2,...,n).

The values of individual diagnostic features j of a stimulant character over time period t for each unit b can be relativized to the other units c according to the formula [43]:

$$Y_{(b/c)jt} = \begin{cases} x_{bjt}/x_{cjt} & x_{cjt} \neq 0 \\ 0 & x_{cjt} = 0 \end{cases}$$ (4)

where: b = 1,2,...,n and c = 1,2,...,n, whereby b≠c, then the correlated values in the period t for each diagnostic feature jin a particular spatial entity in relation to other entities can be presented as relative matrices:

$$\mathbf{Y}_{jt} = \begin{pmatrix} 1 & Y_{(2/1)jt} & \cdots & Y_{(n/1)jt} \\ Y_{(1/2)jt} & 1 & \cdots & Y_{(n/2)jt} \\ \vdots & \vdots & \ddots & \vdots \\ Y_{(1/n)jt} & Y_{(2/n)jt} & \cdots & 1 \end{pmatrix}$$ (5)

Relative matrices $Y\_jt$ are basis for the creation in the period t for each spatial entity i of the matrix Δ_it, which for each entity take the form Lira (2019).

$$\Delta_{1t} = \begin{pmatrix} y_{(1/2)1t} & y_{(1/2)2t} & \cdots & y_{(1/2)mt} \\ y_{(1/3)1t} & y_{(1/3)2t} & \cdots & y_{(1/3)mt} \\ \vdots & \vdots & \vdots & \vdots \\ y_{(1/n)1t} & y_{(1/n)2t} & \cdots & y_{(1/n)mt} \end{pmatrix}$$

$$\Delta_{2t} = \begin{pmatrix} y_{(2/1)1t} & y_{(2/1)2t} & \cdots & y_{(2/1)mt} \\ y_{(2/3)1t} & y_{(2/3)2t} & \cdots & y_{(2/3)mt} \\ \vdots & \vdots & \vdots & \vdots \\ y_{(2/n)1t} & y_{(2/n)2t} & \cdots & y_{(2/n)mt} \end{pmatrix} \tag{6}$$

$$\Delta_{nt} = \begin{pmatrix} y_{(n/1)1t} & y_{(n/1)2t} & \cdots & y_{(n/1)mt} \\ y_{(n/2)1t} & y_{(n/2)2t} & \cdots & y_{(n/2)mt} \\ \vdots & \vdots & \vdots & \vdots \\ y_{(n/n-1)1t} & y_{(n/n-1)2t} & \cdots & y_{(n/n-1)mt} \end{pmatrix}$$

Then, for the data compiled in the matrices $\Delta\_it$, which can be treated as n-1 vectors of observation of the m-featured entities, spatial median was calculated

$$L_1\_med_{it} = (L_1\_med_{i1t}, L_1\_med_{i2t}, \cdots, L_1\_med_{imt})' \tag{7}$$

in time t for each spatial entity i.

The spatial medians became the basis for constructing average relative matrices $\Omega\_t$ in period t:

$$\Omega_t = \begin{pmatrix} \omega_{11t} & \omega_{12t} & \cdots & \omega_{1mt} \\ \omega_{21t} & \omega_{22t} & \cdots & \omega_{2mt} \\ \vdots & \vdots & \vdots & \vdots \\ \omega_{n1t} & \omega_{n2t} & \cdots & \omega_{nmt} \end{pmatrix} \text{ for } t = 1, 2, \ldots, k \tag{8}$$

whereby:for spatial entity i = 1:

$$\omega\_11t = [\![L\_1\_med]\!]\_11t$$

$$\omega\_12t = [\![L\_1\_med]\!]\_12t$$

$$\omega\_1mt = [\![L\_1\_med]\!]\_1mt$$

for spatial entity i = 2:

$$\omega\_21t = [\![L\_1\_med]\!]\_21t$$

$$\omega\_22t = [\![L\_1\_med]\!]\_22t$$

$$\omega\_2mt = [\![L\_1\_med]\!] \_2mt$$

for spatial entity i = n:

$$\omega\_n1t = [\![L\_1\_med]\!] \_n1t$$

$$\omega\_n2t = [\![L\_1\_med]\!] \_n2t$$

$$\omega\_nmt = [\![L\_1\_med]\!] \_nmt.$$

Average relative matrices $\Omega\_t$ are basis for determining the distance from the development pattern of each spatial entity i over time period t. The developmental pattern is in the form of a vector $\omega\_0 = [\omega\_01, \omega\_02, . . ., \omega\_0m]$ ', where $\omega\_oj = \max\_T it\ \{\omega\_ijt\}$, and distances are calculated by taking the wave Hedges metric as a measure of distance:

$$\lambda_{it} = \sum_{j=1}^{m} w_j \left( 1 - \frac{\min(\omega_{ijt}, \omega_{0j})}{\max(\omega_{ijt}, \omega_{0j})} \right) \tag{9}$$

where $w\_j$ are the normed weightings assigned to the j-th feature.

Based on the distance vector $\lambda\_t = [\lambda\_1t, \lambda\_2t, . . ., \lambda\_nt]$ ', relative measures of development $\Phi\_it^{((2))}$ in subsequent periods or moments of time t for individual objects i are constructed:

$$\Phi\_it^{((2))} = \lambda\_it / \mathrm{med}(\lambda\_1t, \lambda\_2t, . . . , \lambda\_nt) \tag{10}$$

where $\mathrm{med}(\lambda\_1t, \lambda\_2t, \cdots, \lambda\_nt)$ means the median of distance for period or moment of time t.

The values of the relative measure $\Phi\_it^{((2))}$ may be greater or less than 1, and its interpretation makes it possible to determine the relative position of the i object in relation to all others. The lower the value of the relative measure is less than 1, the greater is the relative advantage of the object i in relation to others in terms of synthetic evaluation in period or time t. In turn, the greater the value of the measure is than 1, the greater is the relative delay (i.e. the weaker position) of the object i in relation to others.

In the second step, conditions for the development of entrepreneurial capacities were identified. To this end, entrepreneurship index and the density rate of new companies were determined and institutional factors influencing the process of starting a business were analysed.

In the third step, however, statistical analyses were carried out in order to show correlations between the determined relative measures of the economic anchor and selected factors of economic development of the examined Three Seas countries. For this purpose, the Spearman's rank correlations coefficient was applied.

The last element of the research was to estimate the value of economic anchor indicators in 2020 and to outline scenarios for economic development in 3SI countries.

## Results

For the measurement of relative changes in the economic anchor conditions of the EU and 3SI countries in 2015 and 2019, 14 diagnostic features were adopted (Table 2), of which 9 features (X2, X3, X4, X6, X7, X8, X9, X11, X13) were finally selected for testing based on statistical grounds. In the selected group only the X7 variable turned out to be a destimulant, so it was transformed into a stimulant in order to unify the nature of all the selected features (Table 3).

On the basis of the relative values of the synthetic measure of the economic anchor in individual countries, typological classes have been distinguished, bringing together countries with

**Table 3. Basic descriptive statistics characterising the diagnostic features for EU-28 countries.**

| Characteristics | Years | $X_2$ | $X_3$ | $X_4$ | $X_6$ | $X_7$ | $X_8$ | $X_9$ | $X_{11}$ | $X_{13}$ |
|---|---|---|---|---|---|---|---|---|---|---|
| Minimum | 2015 median 2019 | 0.94 0.88 0.63 | 4.51 3.41 3.72 | 8.03 7.26 7.07 | 16.21 16.30 16.35 | 4.60 2.90 2.00 | 63.01 62.42 61.92 | 27.65 30.37 30.01 | 0.14 0.16 0.16 | 40.75 49.20 52.35 |
| Lower quartile | 2015 median 2019 | 2.23 1.94 1.97 | 11.96 11.66 11.13 | 12.83 12.73 12.57 | 72.61 72.05 71.44 | 6.60 4.90 3.88 | 64.83 64.37 63.92 | 40.92 42.41 43.05 | 0.31 0.34 0.38 | 92.98 101.34 106.24 |
| Spatial median | 2015 median 2019 | 12.29 4.35 4.05 | 9.37 12.42 13.51 | 18.38 16.07 15.34 | 82.31 180.81 188.67 | 24.90 10.61 7.25 | 64.57 66.47 65.91 | 76.87 83.36 82.25 | 0.31 0.40 0.42 | 170.18 247.33 246.22 |
| Upper quartile | 2015 median 2019 | 6.06 5.33 4.89 | 19.21 19.25 19.03 | 14.92 14.90 15.08 | 150.65 150.87 151.35 | 10.68 8.78 6.63 | 67.38 66.68 65.91 | 78.62 82.53 83.01 | 0.57 0.60 0.61 | 178.70 199.41 200.28 |
| Maximum | 2015 median 2019 | 23.08 20.28 19.06 | 27.61 28.23 27.82 | 18.43 18.13 17.81 | 1395.84 1461.26 1566.85 | 24.90 21.50 17.30 | 70.73 69.55 69.55 | 221.20 217.62 211.56 | 0.70 0.72 0.71 | 355.50 390.07 394.13 |
| Coefficient of variation (%) | 2015 median 2019 | 91.88 90.95 92.28 | 35.60 38.07 38.52 | 15.44 16.38 16.19 | 149.58 153.46 159.91 | 50.55 53.42 55.15 | 2.89 2.82 2.81 | 61.52 58.60 56.66 | 34.81 34.20 32.06 | 51.43 51.18 50.91 |
| Coefficient of variation based on spatial median (%) | 2015 median 2019 | 69.93 53.65 49.44 | 58.14 32.84 32.25 | 23.06 12.74 11.86 | 47.08 48.63 49.95 | 64.86 43.47 37.91 | 2.42 2.44 2.53 | 37.33 37.58 35.49 | 35.72 32.84 28.35 | 25.54 38.23 39.20 |
| Kurtosis | 2015 median 2019 | 7.80 7.54 7.65 | -0.28– 0.28–0.42 | 1.45 1.74 1.87 | 19.50 19.93 20.68 | 3.42 4.73 5.23 | -0.07 0.03 0.23 | 6.15 5.95 5.65 | -1.02– 1.13–0.94 | 1.22 1.20 1.38 |
| Skewness | 2015 median 2019 | 2.46 2.43 2.45 | 0.28 0.26 0.30 | -0.19– 0.44–0.62 | 4.18 4.23 4.33 | 1.82 1.99 2.12 | 0.45 0.46 0.54 | 2.21 2.14 2.07 | 0.03 0.05– 0.05 | 1.04 1.12 1.20 |

Source: own research.

similar potential level. The comparison of these parameters showed the spatial diversification of the European Union countries (Fig 1).

In the interpretation of the results, it should be noted that the lower the value of the measure is than 1, the greater is the relative advantage of the country in relation to others. In turn, the greater the value of the measure is than 1, the greater is the relative delay, i.e. the weaker is position in relation to others.

Class I, with a relatively high economic anchor value, was formed by countries with a small relative advantage over countries from other groups. Beside Malta, the values determined for this class were between 0.719 and 0.882. There were six countries in the group, including two of the 3SI countries—Austria and Czechia.

Class II is made up of countries with an average indicator level, which in this class took a value from 0.921 to 0.999. This means that the countries included in it had a very small relative advantage over the countries in other classes. It consists of eight countries, including two from the 3SI group—Estonia and Slovenia.

Countries with a low economic anchor ratio formed class III. The values determined for it ranged from 1.001 to 1.088, which indicates a very small relative delay in relation to other groups. This is the most numerous class, which consists of twelve countries, six of which belong to 3SI, these are: Bulgaria, Croatia, Hungary, Latvia, Poland, Slovakia.

The last IV class is made up of only two countries (Lithuania and Romania) with very low economic anchor ratio. Its value ranged between 1.131 and 1.136, which should be interpreted as a small relative delay with respect to the countries from other groups.

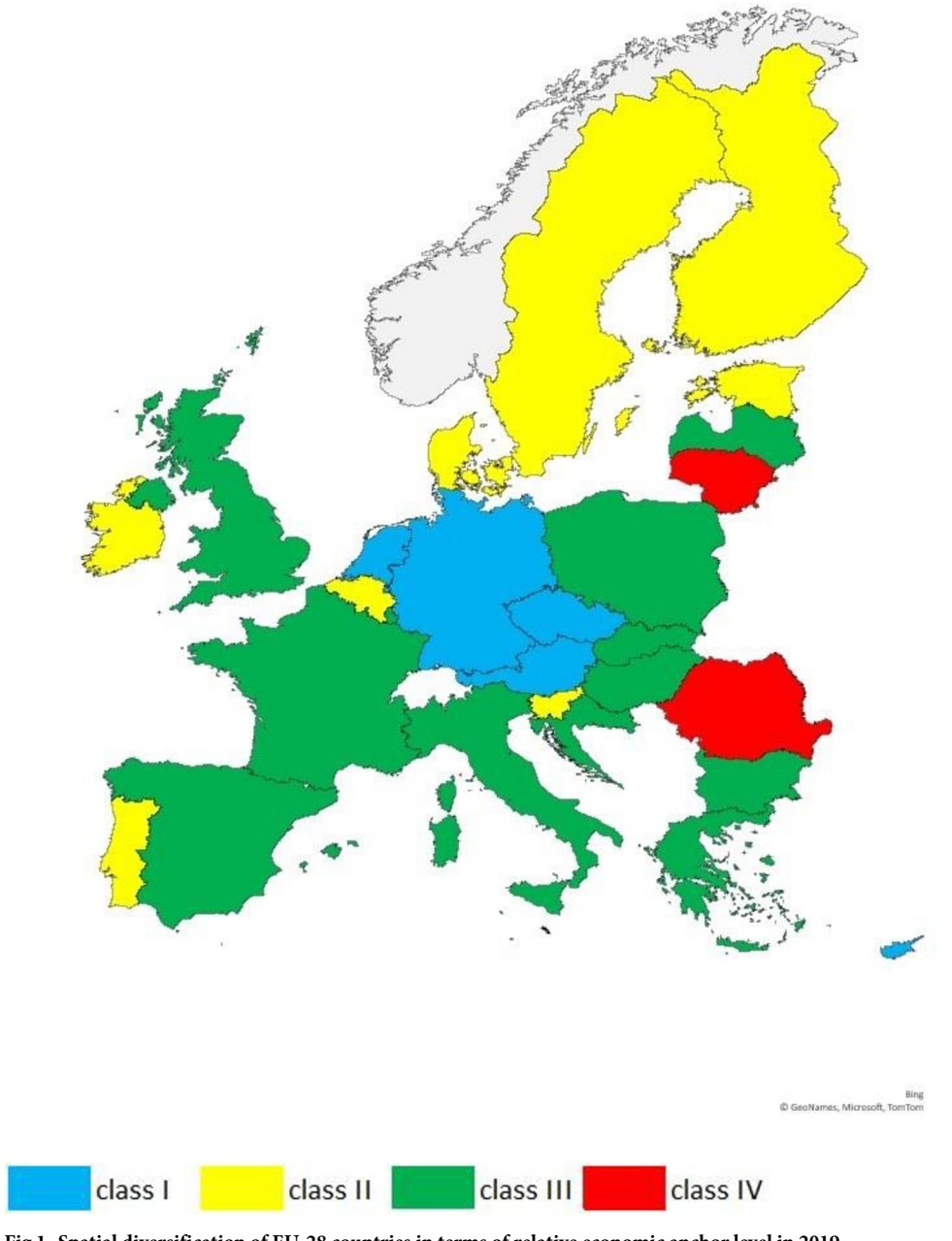

**Fig 1. Spatial diversification of EU-28 countries in terms of relative economic anchor level in 2019.**

In the course of the same procedure, the countries forming 3SI were also assigned to four typological classes (Table 4, Fig 2).

Countries that have achieved an average relative advantage over countries in other classes have formed class I with a high relative economic anchor level (Fig 3). The value of the ratio determined for this class ranged from 0.514 to 0.780 and includes three countries: Austria, Czechia and Slovenia. The values of all diagnostic variables adopted in the study were higher than in other classes. Characteristic for this class was: the lowest level of unemployment, low share of people working in agriculture, high innovation rate and high number of tourists staying overnight (Schneider's index).

**Table 4. Basic descriptive statistics characterising the diagnostic features for 3SI countries.**

| Characteristics | Years | $X_2$ | $X_3$ | $X_4$ | $X_6$ | $X_7$ | $X_8$ | $X_9$ | $X_{11}$ | $X_{13}$ |
|---|---|---|---|---|---|---|---|---|---|---|
| Minimum | 2015 median 2019 | 2.91 2.69 2.62 | 13.10 13.59 12.90 | 12.26 11.55 11.30 | 29.00 29.01 29.22 | 5.10 2.90 2.00 | 65.30 64.42 63.82 | 29.72 30.84 31.45 | 0.14 0.16 0.16 | 40.75 49.20 52.35 |
| Lower quartile | 2015 median 2019 | 3.94 3.45 3.28 | 16.70 16.85 17.53 | 12.71 12.41 12.81 | 59.83 58.87 57.98 | 6.65 4.90 3.78 | 66.40 65.51 64.46 | 48.84 52.61 53.55 | 0.25 0.27 0.29 | 65.88 78.42 85.57 |
| Spatial median | 2015 median 2019 | 8.01 6.78 6.23 | 17.27 17.69 18.13 | 15.08 14.97 14.74 | 64.08 61.68 65.52 | 9.43 7.28 5.47 | 66.95 65.96 65.35 | 61.04 66.83 72.59 | 0.31 0.33 0.35 | 90.88 107.64 110.69 |
| Upper quartile | 2015 median 2019 | 8.40 7.03 6.70 | 22.05 23.08 22.91 | 14.52 14.52 15.14 | 107.11 106.72 106.94 | 9.38 7.35 5.93 | 67.53 66.88 66.27 | 81.45 83.23 84.59 | 0.41 0.41 0.42 | 134.59 157.67 166.48 |
| Maximum | 2015 median 2019 | 23.08 20.28 19.06 | 27.61 28.23 27.82 | 17.16 17.31 16.98 | 133.61 134.12 135.02 | 16.20 11.20 6.60 | 70.73 69.55 68.22 | 121.97 121.04 122.33 | 0.56 0.59 0.60 | 343.49 363.78 381.62 |
| Coefficient of variation (%) | 2015 median 2019 | 73.22 72.55 73.54 | 21.18 27.87 21.93 | 11.89 13.52 12.86 | 42.08 42.69 43.26 | 35.60 35.10 30.62 | 2.30 2.25 2.06 | 36.40 34.97 35.07 | 35.23 38.88 33.03 | 70.18 63.89 62.91 |
| Coefficient of variation based on spatial median (%) | 2015 median 2019 | 40.48 39.59 42.54 | 13.74 15.65 13.05 | 12.52 12.93 12.22 | 56.75 59.07 56.02 | 24.91 22.41 20.11 | 0.91 1.24 1.44 | 30.12 24.61 20.52 | 23.41 21.72 19.52 | 40.44 41.83 38.51 |
| Kurtosis | 2015 median 2019 | 6.05 6.27 6.53 | -0.23– 0.64–0.53 | 0.00– 0.36–0.74 | -1.08–1.14– 1.19 | 2.67 0.90– 0.60 | 1.36 0.72– 0.12 | 0.41 0.09 0.07 | 0.06 0.76 0.44 | 4.34 3.49 3.88 |
| Skewness | 2015 median 2019 | 2.26 2.31 2.37 | 0.40 0.33 0.31 | 0.83 0.51 0.29 | -0.39–0.37– 0.34 | 1.44 0.75– 0.09 | 1.15 0.88 0.69 | 0.38 0.15 0.18 | 0.49 0.63 0.50 | 1.95 1.74 1.80 |

Source: own research.

Class II is made up of countries with an average relative level of development. The calculated index was between 0.926 and 1.000, which means that countries in this group had a very small relative advantage over the countries in other groups. Estonia, Hungary, Poland and Slovakia were included in this group. This class was characterised by a low level of unemployment below 5%, a relatively high innovation coefficient in relation to Class I, a lower share of exports of goods and services in GDP in relation to Class I and III.

Countries with a low economic anchor ratio formed class III. The determined values ranged from 1,096 to 1,157, which indicates a small relative delay in relation to other groups. The group consists of Bulgaria, Croatia and Latvia. In this class there was an increase in the share of employed in agriculture, a significant decrease in the share of employed in industry, which resulted in a decrease in the innovation rate. However, they recorded the highest share of exports of goods and services in GDP compared to other classes. A high rate of tourist traffic intensity-Schneider's index was also noted, as the countries that make it up are the destinations of choice for tourists from Europe and beyond.

The last IV class consists of Romania and Lithuania, with a relatively very low AE level—from 1.201 to 1.313, which should be read as an average relative delay compared to the countries in other classes. The values of all the diagnostic variables adopted in the study for these two countries were lower than for other classes. A high share of people working in agriculture, and a low share of people working in industry and trade and a low share of exports of goods and services in GDP, characterised this group.

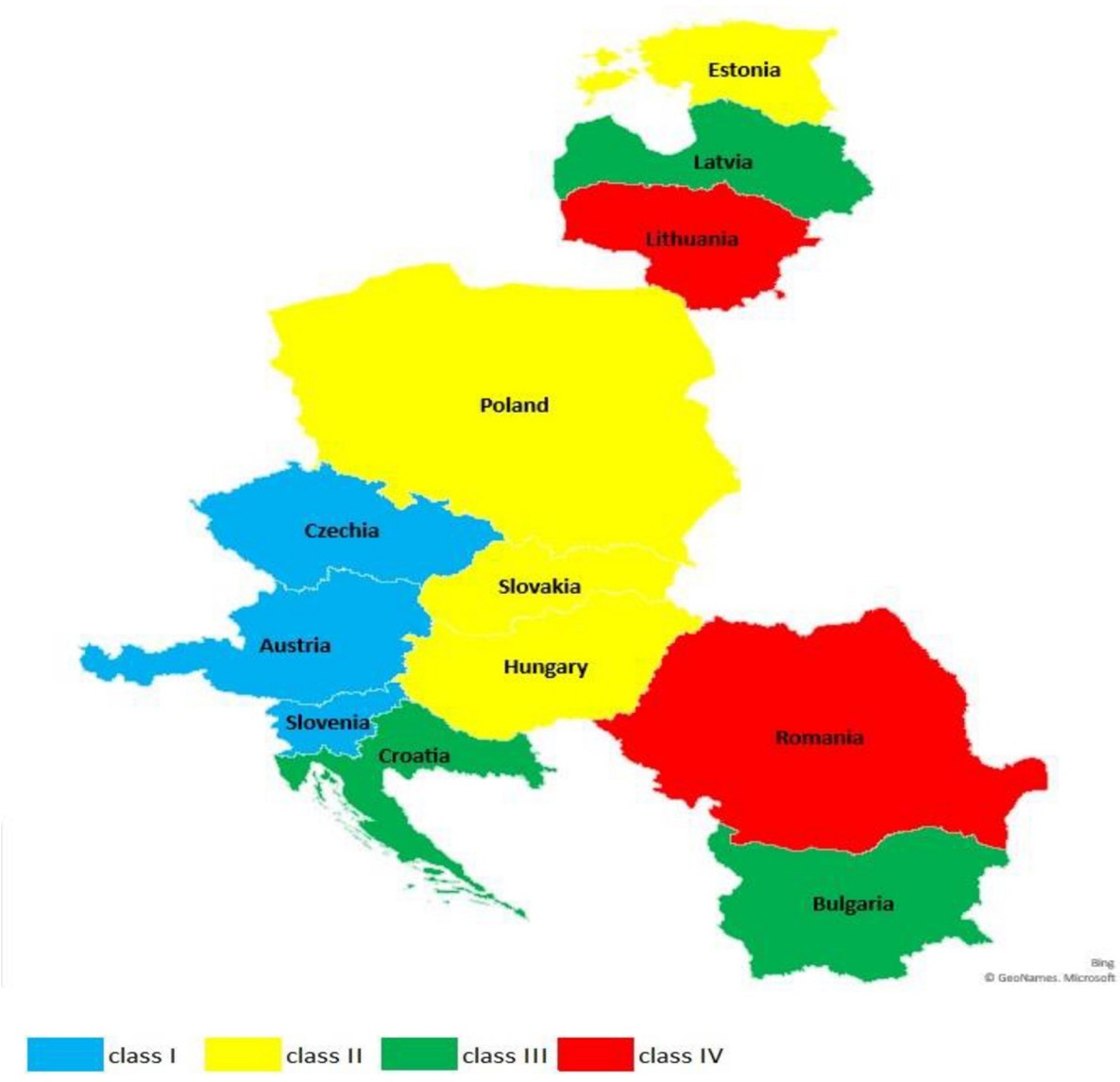

**Fig 2. Spatial diversification of 3SI countries in terms of relative economic anchor level in 2019.**

Between 2015 and 2019, the largest relative advantage in the 3SI group was achieved by Austria, with two other countries, Czechia and Slovenia, having an average relative advantage. Czechia, Slovakia and Latvia reduced the relative delay to the largest extent, and the largest deterioration of the economic situation in relation to the other 3SI countries was observed in Bulgaria, Croatia, Lithuania, Estonia (Fig 3).

In order to explain the demonstrated differences in the level of the 3SI countries' economic anchor ratio, the determinants influencing the development of the level of entrepreneurship in these countries were analysed. Many institutions monitoring the European market carry out this type of research. The most important of these include the World Bank, OECD and European Commission. It has been assumed that, similarly to the World Bank's analyses, entrepreneurship expresses the process of setting up and running a business [44–46]. The development of entrepreneurship is a key factor in economic development, and its high level influences the rate of GDP growth and has a positive impact on the labour market, leading to

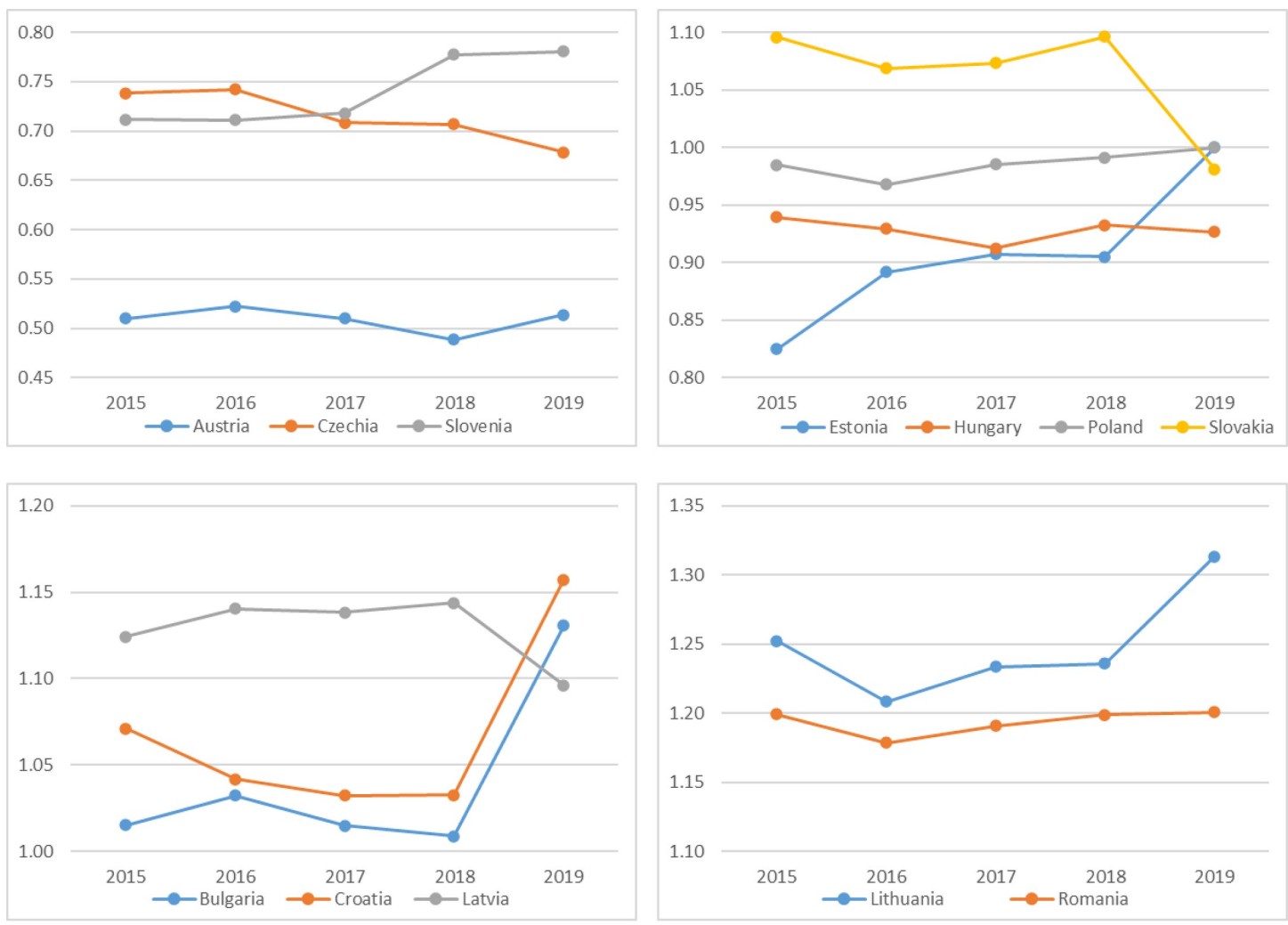

**Fig 3. Changes in the relative levels of the 3SI countries' economic anchors between 2015–2019.**

an increase in employment. The countries of the European Union, including the 3SI countries, differ significantly in terms of the degree of creation of entrepreneurial attitudes. This may be due to many factors of both economic and social nature. Possible statistical data showing the number of companies registered in EU countries cover the years 2015–2017. On the basis of these findings, it was established that in the EU-28 countries, the business activity was carried out in 2017 by more than 24.3 million companies, including 5.6 million in 3SI countries (23.2%). It is worth noting that the average annual growth rate of the number of enterprises in the EU-28 countries was 1.9%, and in 3SI countries—3.0%. (Fig 4). The largest number of registered entities is located in countries with well-developed market economies, i.e. Italy, France, Spain, Germany and the United Kingdom. On the other hand, in the 3SI group of countries the highest number of entities was recorded in Poland, Czechia and Hungary.

In order to compare the diversity of entrepreneurial activity, the number of companies per 1,000 inhabitants was calculated (entrepreneurship index) (Fig 5). This made it possible to determine the actual entrepreneurial potential, the level of which is a function of the entrepreneurial attitudes of the country's inhabitants. The highest values of the index were recorded in Czechia, Slovakia and Portugal. The lowest in countries such as Romania, Germany or the

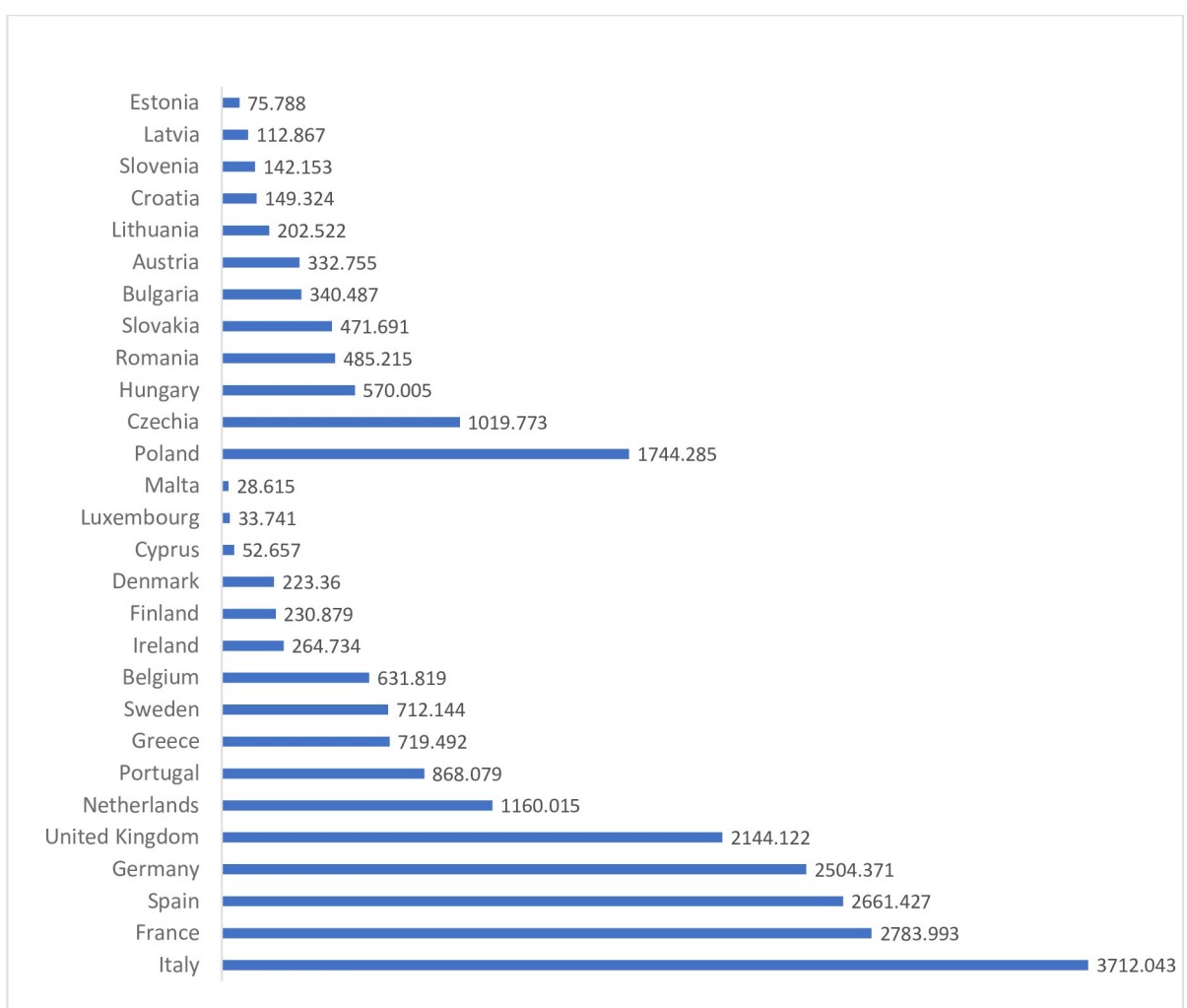

**Fig 4. Number of companies in EU-28 and 3SI in 2017 (in thousands).**

United Kingdom. Comparing data on the number of companies with the number of companies per 1,000 inhabitants, it is clear that the capacities for the development of entrepreneurship in countries such as Poland, Hungary and Romania have not yet been used.

Stimulating the entrepreneurial attitudes of the inhabitants of a given country is an important task of economic policy, which means that many determinants of the development of entrepreneurship are dependent on state policy. This applies primarily to factors of an institutional and formal-legal nature having, inter alia, a significant impact on: the time that an entrepreneur has to devote to register its business, the number and transparency of procedures related to the registration of business activity, the amount of costs associated with starting up a business, or, for example, the waiting time for a construction permit (Table 5).

Analysing the data, it can be seen that the fastest registration of activity is in the Baltic States (Estonia, Latvia, Lithuania) and in Hungary, Slovenia. However, the longest waiting time to start business is in Romania (35 days) and Poland (37 days). An important aspect of starting a business activity is also the costs of its creation, calculated as a percentage of national income per capita. No costs for the creation of a new project are borne by the inhabitants of Slovenia, small payments are made in Romania (0.4%), Lithuanians (0.5%) and the Czechia and Slovaks

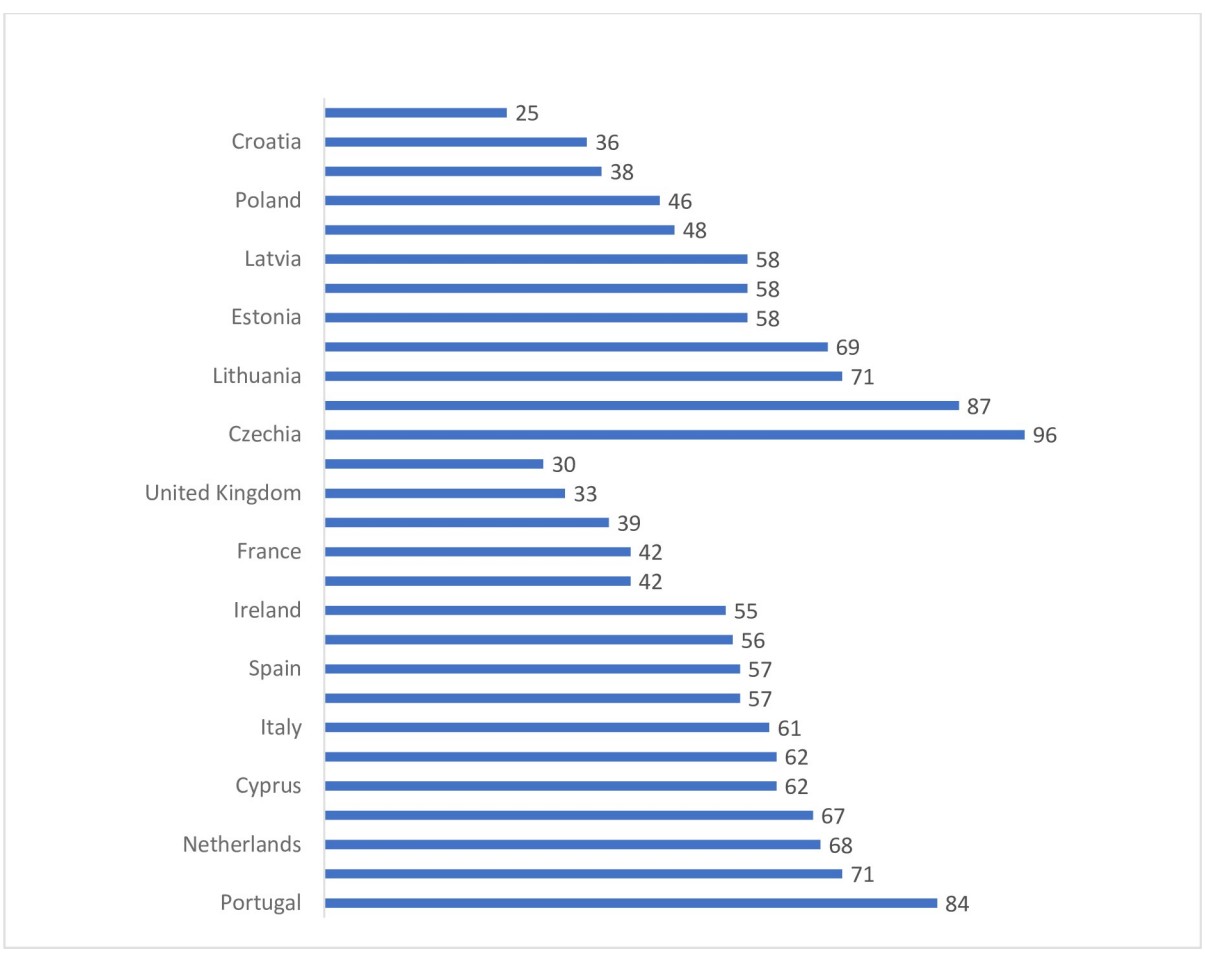

**Fig 5. Entrepreneurship index in EU-28 and 3SI in 2017.**

(1.0% of national income). The most expensive is to establish a company in Poland, Croatia, Hungary and Austria.

From the point of view of the authors of this study, an important factor for conducting business activity is the number of days awaiting the construction permit. The development of new activities often requires a construction of a production hall or facility in which production or services will be provided. The longest waiting time for a permit is in Slovakia, Romania, Slovenia and Czechia. On the other hand, the shortest time is in Lithuania and Bulgaria. Taking into account the data on the density of new business (i.e. the number of new registrations per 1,000 persons aged 15–64), it should be noted that in 3SI countries there is a very high diversity in this respect, i.e. the highest average values were recorded in Estonia, Bulgaria, Latvia and Romania and the lowest values in Austria and Poland. Undoubtedly, the establishment of new entities was influenced by the previously discussed conditions related to the commencement of business activity, and this assumption was confirmed by an analysis of the correlation of Spearman's rank (Table 6). It showed a negative correlation of moderate union strength, for the number of days needed to start a business ($r_{2018} = -0.339$) and for the number of days awaiting a construction permit, the correlation showed a higher relation strength ($r_{2017} = -0.483$).

The next stage of the research was to determine the correlation between the values of the synthetic economic anchor measure and the factors that indicate the economic development

**Table 5. Analysed conditions for doing business in 3SI countries.**

| Country | Number of days for starting business | | | Number of procedures for starting business | | | Costs of starting business (% of income per capita) | | | Number of days awaiting the construction permit | | |
|---|---|---|---|---|---|---|---|---|---|---|---|---|
| | 2015 | 2017 | 2019 | 2015 | 2017 | 2019 | 2015 | 2017 | 2019 | 2015 | 2017 | 2019 |
| Austria | 22 | 21 | 21 | 8 | 8 | 8 | 0.3 | 0.3 | 4.8 | 192 | 222 | 222 |
| Bulgaria | 18 | 23 | 23 | 4 | 6 | 7 | 0.8 | 1.3 | 1.1 | 110 | 130 | 97 |
| Croatia | 15 | 7 | 22.5 | 7 | 8 | 8 | 3.5 | 7.3 | 6.6 | 188 | 127 | 146 |
| Czechia | 19 | 9 | 24.5 | 9 | 8 | 8 | 8 | 5.7 | 1.0 | 143 | 247 | 246 |
| Estonia | 4.5 | 3.5 | 3.5 | 4 | 3 | 3 | 1.4 | 1.2 | 1.1 | 103 | 102 | 103 |
| Hungary | 5 | 7 | 7 | 4 | 6 | 6 | 8.3 | 7.1 | 4.9 | 91 | 202 | 192.5 |
| Latvia | 12.5 | 5.5 | 5.5 | 4 | 4 | 4 | 3.6 | 1.5 | 1.6 | 179 | 147 | 192 |
| Lithuania | 3.5 | 5.5 | 5.5 | 3 | 4 | 4 | 0.7 | 0.6 | 0.5 | 91 | 103 | 74 |
| Poland | 30 | 37 | 37 | 4 | 4 | 5 | 12.9 | 12.1 | 11.8 | 212 | 153 | 153 |
| Romania | 8 | 12 | 35 | 5 | 6 | 6 | 2.1 | 2 | 0.4 | 255 | 171 | 260 |
| Slovakia | 11.5 | 11.5 | 26.5 | 7 | 6 | 8 | 1.5 | 1.2 | 1.0 | 286 | 286 | 300 |
| Slovenia | 6 | 7 | 8 | 2 | 4 | 3 | 0 | 0 | 0 | 212.5 | 224.5 | 247.5 |

Source: own research based on [44–46].

of the countries concerned, i.e. the number of enterprises per 1,000 inhabitants, the density of new companies and the level of GDP per capita. The calculated values of the Spearman's correlations coefficient indicate a positive correlation with a moderate relationship in the case of the number of companies per 1,000 inhabitants, which confirms that in the countries with a higher economic anchor there are more economic entities. In contrast, in the case of a density of new businesses, the correlation was negative with a moderately high strength of the relation, which means that more new business entities are emerging in relatively delayed countries. Undoubtedly, the indicator of whether a country is developing or not is GDP per capita. This indicator is often used in international analyses as it shows the level of development of the countries being compared. The calculated correlation coefficients showed a positive, statistically highly significant relations with high strength of the relations.

**Table 6. New business density in 3SI countries.**

| Country | New registrations per 1,000 people ages 15–64 | | | | |
|---|---|---|---|---|---|
| | 2015 | 2016 | 2017 | 2018 | Average |
| Austria | 0.63 | 0.61 | 0.63 | 0.65 | 0.63 |
| Bulgaria | 9.86 | 10.91 | 10.83 | 10.10 | 10.43 |
| Croatia | 4.51 | 4.90 | 5.52 | 5.86 | 5.20 |
| Czechia | 3.70 | 3.98 | 4.49 | 4.39 | 4.14 |
| Estonia | 17.54 | 19.43 | 21.74 | 23.59 | 20.58 |
| Hungary | 3.14 | 3.36 | 3.47 | 3.74 | 3.43 |
| Latvia | 9.71 | 8.09 | 7.45 | 8.01 | 8.31 |
| Lithuania | 3.19 | 3.34 | 3.53 | 3.33 | 3.34 |
| Poland | 1.53 | 1.66 | 1.50 | 1.44 | 1.53 |
| Romania | 4.82 | 5.58 | 7.54 | 7.32 | 6.31 |
| Slovakia | 2.72 | 4.72 | 5.24 | 5.25 | 4.48 |
| Slovenia | 4.00 | 3.13 | 3.34 | 3.09 | 3.39 |

Source: own research.

**Table 7. Spearman's correlations coefficient values for the synthetic economic anchor measure and the entrepreneurship index, New business density, Gross domestic product *per capita* and The Global Competitiveness Index.**

| Specification | Years | | | | |
|---|---|---|---|---|---|
| | **2015** | **2016** | **2017** | **2018** | **2019** |
| Synthetic relative measure vs. | | | | | |
| Number of companies per 1,000 inhabitants | 0.343 | 0.364 | 0.349 | × | × |
| New business density | -0.315 | -0.427 | -0.510 | -0.434 | × |
| Gross domestic product *per capita* | 0.783* | 0.692* | 0.748* | 0.720* | 0.629* |
| The Global Competitiveness Index | × | × | 0.706* | 0.706* | 0.655* |

× no dat

* p<0.01.

Source: own research.

These results confirm the accuracy of the selection of features for the construction of the synthetic economic anchor ratio. This is also documented by the correlation analysis with the Global Competitiveness Index. Its results indicate a positive highly significant correlation with a moderately strong compound (Table 7).

The final stage of the research was to determine the probable consequences of a coronavirus epidemic on the economic situation. Due to a significant degree of unpredictability, the forecast is of a short-term nature and concerns the second half of 2020. However, this does not mean that the assumed prediction time determines the return to the so-called normality before COVID-19. It is, however, an indication of possible future changes. For this purpose, the 3-element simple and weighted moving average method was used. (Table 8, Fig 6). The best forecast model was selected using the least ex-post error criterion for expired forecasts. The model was selected on the basis of the mean absolute percentage error (MAPE).

The projection indicates that six countries, i.e. Austria, Czechia, Slovenia, Hungary, Bulgaria and Lithuania, may see a reduction in the value of the economic anchor ratio, which means reduction in disparities, including the largest for Slovenia. The index values for Poland and Estonia remain unchanged. For other 3SI countries, on the other hand, the indicator is expected to increase, which means increase in disparities, i.e. increase in relative delay. However, it should be stressed that the processes of restoring the entrepreneurial potential of

**Table 8. Synthetic economic anchor measure for 2020.**

| Country | Average 3-element simple | MAPE (%) | Average weighted 3-element (weights 0.2; 0.3; 0.5) | MAPE (%) |
|---|---|---|---|---|
| Austria | 0.510 | 2.45 | 0.510 | 2.44 |
| Bulgaria | 1.126 | 1.36 | 1.129 | 1.25 |
| Croatia | 1.162 | 0.96 | 1.161 | 0.84 |
| Czechia | 0.667 | 2.17 | 0.669 | 2.19 |
| Estonia | 1.000 | 1.88 | 1.000 | 1.20 |
| Hungary | 0.915 | 1.21 | 0.919 | 1.32 |
| Latvia | 1.099 | 0.89 | 1.098 | 0.77 |
| Lithuania | 1.301 | 1.42 | 1.305 | 1.27 |
| Poland | 1.000 | 0.93 | 1.000 | 0.62 |
| Romania | 1.203 | 0.76 | 1.203 | 0.67 |
| Slovakia | 0.983 | 0.98 | 0.983 | 0.74 |
| Slovenia | 0.759 | 6.54 | 0.767 | 5.83 |

Source: own research.

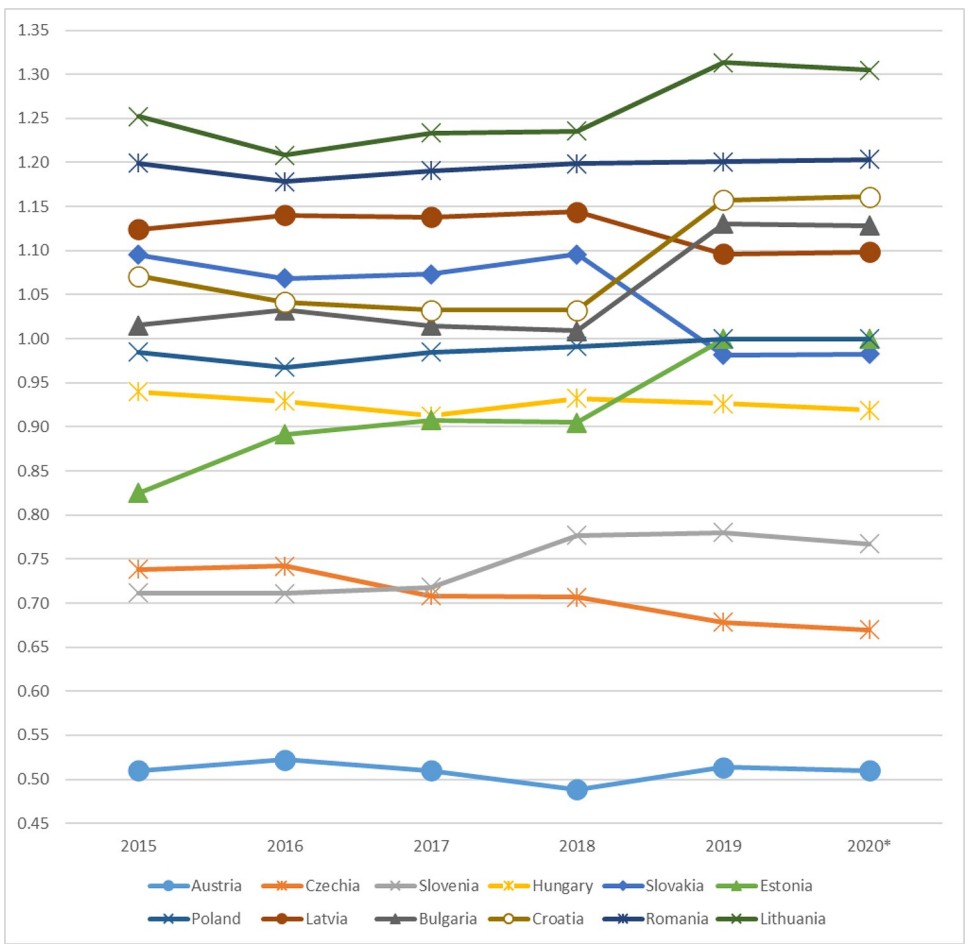

**Fig 6. 3SI synthetic anchor ratio projection for the second half of 2020 (3-element weighted average).**

individual countries will depend on the choices made by public authorities. It is considered that decisions targeted at new solutions, requiring time and resources for investment, can clearly slow down the process of economic rebalancing after COVID-19. It is therefore recommended to implement previous pathways of action and restoration of the already existing capacity [14]. It seems that most 3SI countries should take this course of action.

## Discussion

Due to the relatively short period of formal functioning of the research entity—a group of countries of the Three Seas Initiative (3SI) and the pioneering approach of the research subject—the spatial distribution of entrepreneurial potential. It is also difficult to find a direct reference point for the findings made and to verify them. Even less research concerns the spatial diversification of observed and possible future manifestations and effects of the eco-nomic shock caused by the COVID-19 pandemic. In our opinion, its occurrence and potential effects have become a kind of test of the efficiency and effectiveness of the entrepreneurial potential and its ability to overcome the economic lockdown. As Peter Drucker wrote, "Trying to predict the future is like trying to drive down a country road at night with no lights while looking out the back window", it is not easy to predict, in the absence of a full recognition of the effects of

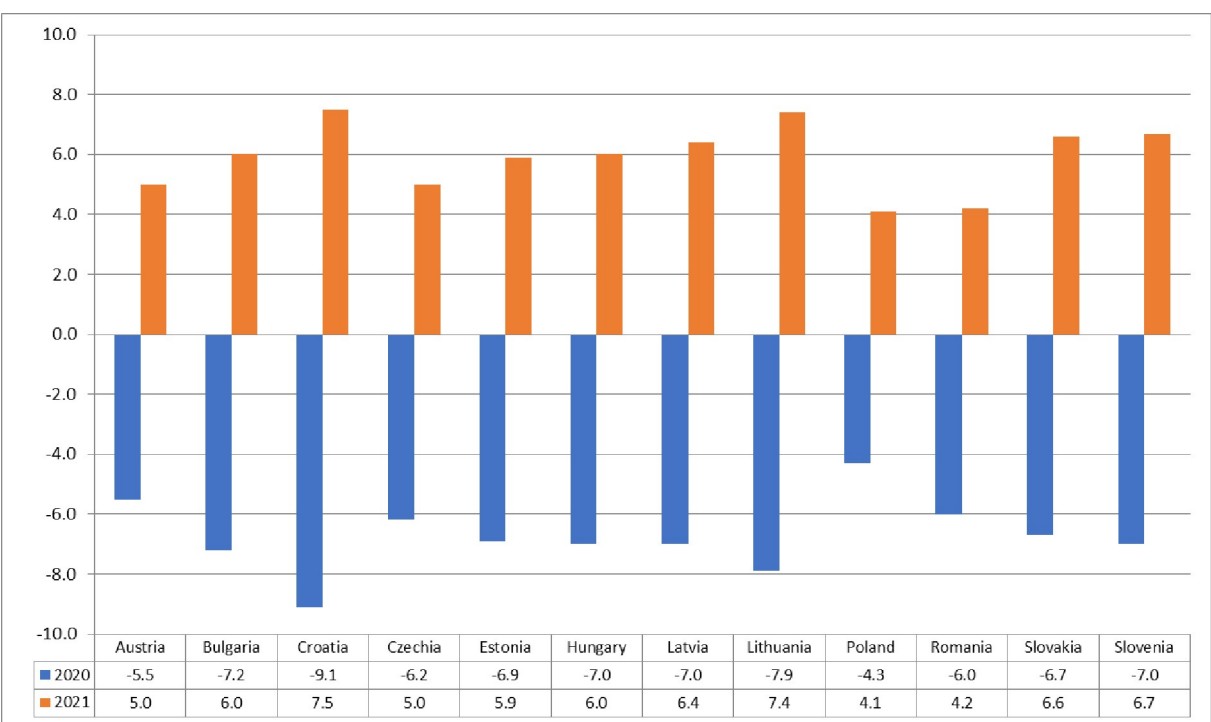

**Fig 7. GDP forecast for 2020 and 2021 (%).** Source: own elaboration based on the European Commission's forecast.

the pandemic, how long it will take to commission the frozen socio-economic potential for several months. However, it is possible to assume a positive scenario in which, thanks to the support provided by the EU funds earmarked for the pandemic, it will be possible to quickly activate the business potential and rebuild the economies of individual 3SI countries. The European Commission has proposed in May this year a new recovery instrument, Next Generation EU, which will provide more than € 750 billion to countries affected by COVID-19, out of which €187.7 billion is allocated to 3SI countries. It is worth nothing that the largest beneficiaries of this aid will be Poland and Romania, which together will have more than 50% of the funds at their disposal (the median for the Three Seas group is €11.2 billion). In terms of the EU as a whole, the largest stream of the European aid will flow to Italy and Spain, the countries most affected by the COVID-19 pandemic. The lack of external financial transfers to the 3SI countries may result in a negative image of these countries defined as a "specific area of structural negligence" [7]. Such poorly optimistic conclusions are led, among others, by the economic recession fore-casts for the second part of 2020 and for 2021, published by the European Commission and covering all EU countries (Fig 7). They are very diverse, and their conclusion is not optimistic. A deep and uneven recession and an uncertain economic recovery will affect all EU countries [47, 48].

The outbreak of the viral disease known as COVID-19 was reported in Europe in January 2020. At the time, no one predicted the speed of its development or that it would be declared a pandemic three months later. Its effects have had an unforeseen impact on the socio-economic condition of economies around the world, including the countries making up the Three Seas Initiative. The implementation of the lockdown measures associated with COVID-19, although justified in terms of limiting the spread of the pandemic, led to a state of economic hibernation. The closure of enterprises, restrictions on the functioning of the supply chain or

the enforced temporary closure of many industries, including services, resulted in an unprecedented fall in economic activity. It is worth emphasising that the restrictions introduced in most countries resulted in a reduction of employment and, consequently, an increase in unemployment. Despite initially negative forecasts, the pace of this phenomenon was moderate [49, 50].

The relatively mild response of the labour market (measured by the unemployment rate) was largely due to the change in the form of work from stationary to remote and the concomitant reduction in working time (measured by the number of hours worked). Business meetings have also moved to the virtual zone (e.g. Zoom, etc.). The forced remoteness in many cases contributed to the acquisition of a new competence, i.e. the ability to communicate with clients from all over the world [51].

The time of the pandemic revealed variations in the share of remote work in relation to traditional forms of performing work in individual Three Seas Initiative countries. The least effective in this respect were Romania, Hungary, Croatia, Poland, Slovakia and Czechia. In Austria, on the other hand, the share was higher than the EU average. This was caused, on the one hand, by the possessed ICT infrastructure, and on the other hand, by the organisational structure of particular segments of these countries' economies [52]. In countries with a significant share of SME enterprises, the share of remote working was relatively lower.

According to Gębska M. [53], the Three Seas Initiative countries face many challenges in recovering from the COVID-19 pandemic. Among nothing mentioned are difficulties in increasing production, caused by the possibility of further waves of the pandemic, the need to increase government spending on the health sector, or support for enterprises weakened by the pandemic. Identifying the entrepreneurial potential of individual countries may prove helpful in initiating the recovery of economies at the level of microactivity, which tends to stimulate consumer demand and boost the economy as a whole.

The COVID-19 pandemic will also test the sustainability of the Three Seas Initiative. While a joint financial facility has been set up to support investment within the 3SI, so far no targets have been agreed for its disbursement, nor has any joint action been taken to mitigate the impact of the COVID-19 pandemic.

The recovery of individual countries' economies will therefore depend not only on the course and effects of the pandemic, but above all on the sustainability of their economic structures, including their entrepreneurial potential. An additional factor that will influence the pace of the de-freezing the economy will also be the numerous interdependencies of EU economies, which can have a knock-on effect on other Member States.

An important sector of the economy of the analysed countries, especially for the three seaside countries, is tourism. Certainly, the tourism sector was most affected by the coronavirus pandemic. The countries with the highest share of the tourism sector in GDP are Croatia, Austria, Estonia, Slovenia and Bulgaria [54].

## Conclusions

The entrepreneurial potential of a country or groups of countries is one of the elements that describe the internal possibilities of creating progress and development of a country. Its final level is influenced by internal and external factors. An example of the impact of the latter is the outbreak of the coronavirus pandemic, which caused a shock to economic systems and their temporary "dormancy". Bearing this in mind, the authors have sought to diagnose the entrepreneurial potential of the 3SI countries in the context of assessing their ability to quickly restart after the lockdown.

The study found that the entrepreneurial potential of the 3SI countries in 2015–2019 were determined by variables related to six areas, namely, i.e. shares of employees in agriculture,

forestry and fishing, manufacturing and trade (local economy), population density (demographic situation), shares of the unemployed in the total number of active people and people of working age -15-64 years (social situation), export of goods and services in % of GDP (trade exchange), innovation index (innovation) and tourist traffic intensity-Schneider's index (tourism economy), on the basis of which the taxonomic measure of the economic anchor (synthetic measure of the economic anchor) was built.

Moreover, a very large spatial diversification in the value of this indicator was indicated. The results allowed to conclude that large differences in entrepreneurial potential (measured by the economic anchor ratio) exist not only within the 3SI countries, but are also visible in the EU-28.

Analysing the calculated values of the entrepreneurship index, it was found that the highest levels of the calculated values were recorded in the countries belonging to the 3SI: Czechia, Slovakia and Lithuania, the lowest values were observed in Romania, Croatia and Austria. It is worth noting that in the EU-28 the highest entrepreneurship index, at a similar level to 3SI, was recorded in Portugal and Sweden. Approximately the lowest values in Germany and the UK. At the same time, it was found (by comparing data specifying the total number of economic entities with the number of companies per 1,000 inhabitants) that entrepreneurial potential has not been used so far in countries such as Poland, Hungary and Romania. This means that not in all 3SI countries changes in the defined factors shaping entrepreneurial potential are conducive to its development.

Macroeconomic projections are almost always subject to major errors, especially if the shock that caused the crisis comes from outside the economic system, as in the case of the COVID-19 pandemic. Nevertheless, the observed changes in the level of entrepreneurial potential of the 3SI countries and forecasts prepared by the European Commission should be considered as a warning against underestimating the scale of negative effects of COVID-19 on the entrepreneurial potential of the twelve countries forming the Three Seas Initiative.

In conclusion, the study has several limitations that could be further explored in future research. The main limitation of the research is no statistical data from 2020 and 2021 on the development of the economies of the Three Seas countries. Hence, data on government support counteracting the COVID-19 pandemic should be analysed. Therefore, future research should focus on this aspect.

## Supporting information

**S1 Table.**
(DOCX)

## Author Contributions

**Conceptualization:** Magdalena Kozera-Kowalska, Jarosław Lira.

**Data curation:** Jarosław Uglis.

**Formal analysis:** Jarosław Uglis.

**Investigation:** Magdalena Kozera-Kowalska.

**Methodology:** Jarosław Uglis, Jarosław Lira.

**Software:** Jarosław Lira.

**Writing – original draft:** Magdalena Kozera-Kowalska.

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
