## [Decision Letter · Decision Letter 0]

3 May 2021

PONE-D-21-02343

A framework to measure the taxonomic of economic anchor: a case study of the Three Seas Initiative countries

PLOS ONE

Dear Dr. Magdalena Kozera-Kowalska,

Thank you for submitting your manuscript to PLOS ONE. After careful consideration, we feel that it has merit but does not fully meet PLOS ONE’s publication criteria as it currently stands. Therefore, we invite you to submit a revised version of the manuscript that addresses the points raised during the review process.

We look forward to receiving your revised manuscript.

Kind regards,

Carlos Alberto Zúniga-González, Ph.D

Academic Editor

PLOS ONE

Additional Editor Comments:

Dear authors, I request that the improvements indicated by the reviewers be made in order to improve the quality of the same in a priority issue for our society.

Journal Requirements:

3. We note that Figures 1 and 2 in your submission contain map images which may be copyrighted. All PLOS content is published under the Creative Commons Attribution License (CC BY 4.0), which means that the manuscript, images, and Supporting Information files will be freely available online, and any third party is permitted to access, download, copy, distribute, and use these materials in any way, even commercially, with proper attribution. For these reasons, we cannot publish previously copyrighted maps or satellite images created using proprietary data, such as Google software (Google Maps, Street View, and Earth). For more information, see our copyright guidelines: http://journals.plos.org/plosone/s/licenses-and-copyright.

3.1.    You may seek permission from the original copyright holder of Figures 1 and 2 to publish the content specifically under the CC BY 4.0 license. 

3.2.    If you are unable to obtain permission from the original copyright holder to publish these figures under the CC BY 4.0 license or if the copyright holder’s requirements are incompatible with the CC BY 4.0 license, please either i) remove the figure or ii) supply a replacement figure that complies with the CC BY 4.0 license. Please check copyright information on all replacement figures and update the figure caption with source information. If applicable, please specify in the figure caption text when a figure is similar but not identical to the original image and is therefore for illustrative purposes only.

Reviewers' comments:

Reviewer's Responses to Questions

**Comments to the Author**

1. Is the manuscript technically sound, and do the data support the conclusions?

Reviewer #1: Yes

Reviewer #2: Yes

2. Has the statistical analysis been performed appropriately and rigorously? 

Reviewer #1: Yes

Reviewer #2: Yes

3. Have the authors made all data underlying the findings in their manuscript fully available?

Reviewer #1: Yes

Reviewer #2: Yes

4. Is the manuscript presented in an intelligible fashion and written in standard English?

Reviewer #1: Yes

Reviewer #2: Yes

5. Review Comments to the Author

Reviewer #1: The paper is interesting. However, I believe there are several indexes available that partially measures the entrepreneurial potential of countries. For instance, the world bank has "Ease of doing business index". In fact, the doingbusiness project of the World Bank incorporates many of the indicators that might be helpful for the discussion. The authors may wish to discuss how their measure differs from the other alternative measures, and why their measure is important.

The paper mentions that the measure developed in the study can help improve "the ability to overcome the consequences of extraordinary events, such as COVID-19 and prospects for the return to an accelerated development once the destabiliser of the economic system." However, I think the discussion section of the paper, although focuses on Covid-19, is only weakly correlated with the results of the paper. I think the authors ought to strengthen the discussion section, and further the linkage between the analysis conducted in the paper and Covid-19.

Reviewer #2: The authors have selected an important dimension and presented the paper paper in relatively easy to read, for a general reader, manner. However, the study cannot be linked to COVID-19. COVID-19 was first case was registered in China in the last quarter of 2019. In the selected countries it spread afterwards. So, the impact of COVID cannot be determined. For this, additional data of 2020 is required and it could be a much better before after situation analyses.

6. PLOS authors have the option to publish the peer review history of their article (what does this mean?). If published, this will include your full peer review and any attached files.

Reviewer #1: No

Reviewer #2: No

---

## [Author Response · Author response to Decision Letter 0]

12 May 2021

I want to informe that:

There is no conflicts of interest associated with this publication, and there has been no significant financial support for this work that could have influenced its outcome – so “the authors received no specific funding for this work.”

I would also like to inform you that Figures 1 and 2 - are graphical representations of our research and were made by us using Excel Microsoft 365 (the Bing GeoNames.Microsoft tool). Each figure contains the designation of this program in the bottom right corner. As the authors, we are therefore the copyright holders of these figures and transfer them to Plos-One.

Response to reviewesr:

Thank you for giving us the opportunity to submit a revised draft of the manuscript “A framework to measure the taxonomic of economic anchor: a case study of the Three Seas Initiative countries” for publication in the PLOS ONE.

We appreciate the time and effort that you and the reviewers dedicated to providing feedback on our manuscript and are grateful for the insightful comments on and valuable improvements to our paper. We have incorporated all the suggestions made by the reviewers. Those changes are highlighted within the manuscript.

Please find our detailed responses below:

Reviewer #1:

1: The paper is interesting. However, I believe there are several indexes available that partially measures the entrepreneurial potential of countries. For instance, the world bank has "Ease of doing business index". In fact, the doing business project of the World Bank incorporates many of the indicators that might be helpful for the discussion.

The authors may wish to discuss how their measure differs from the other alternative measures, and why their measure is important.

Author response: Thank you for your suggestion on this very essential point.

We agree with the above comment that there are several indicators available that partially measure the entrepreneurial potential of countries. These include those listed in the annual Doing Business report - the flagship publication of the World Bank Group. Doing Business includes 11 indicators to measure aspects of business regulation that are relevant to entrepreneurship. We used 4 of these in our work (they are included in Table 7).

In designing our economic anchor indicator, we aimed to create an indicator that reflects as fully as possible the factors that influence the development of entrepreneurship. To this end, 14 diagnostic features were selected, of which, after preliminary statistical analysis, 9 were used. In our opinion, the selected 9 features constitute the necessary basis for entrepreneurship development.

In addition, a relative pattern method based on Weber's median was used to develop an innovative economic anchor index. The advantage of this indicator is the easy interpretation of the obtained results, since the values of the relative measure Φ_it^((2)) can be greater or less than 1, which makes it possible to determine the relative position of an object, in this case countries, in relation to all others (lines 262-267).

2: The paper mentions that the measure developed in the study can help improve "the ability to overcome the consequences of extraordinary events, such as COVID-19 and prospects for the return to an accelerated development once the destabiliser of the economic system", However, I think the discussion section of the paper, although focuses on Covid-19, is only weakly correlated with the results of the paper. I think the authors ought to strengthen the discussion section, and further the linkage between the analysis conducted in the paper and Covid-19.

Author response: Thank you for your kindly suggestion.

Due to the lack of available data on the economic situation of individual countries in the period under review, it was not possible to deepen the analysis during the preparation of the text. Nevertheless, in accordance with the reviewer's suggestion, the discussion part of the article was strengthened in an attempt to clarify the relationships. (line 507 to 541).

Reviewer #2

1: The authors have selected an important dimension and presented the paper in relatively easy to read, for a general reader, manner. However, the study cannot be linked to COVID-19. COVID-19 was first case was registered in China in the last quarter of 2019. In the selected countries it spread afterwards. So, the impact of COVID cannot be determined. For this, additional data of 2020 is required and it could be a much better before after situation analyses.

Author response: Thank you for your kindly suggestion.

The reviewer's observation is correct. The first official case of COVID-19 in Europe was reported on 24 January 2020. The virus, which causes severe infectious disease, spread rapidly across the continent, and on 11 March the World Health Organisation declared a global pandemic. We wrote the article in July-October 2020, without yet having data on the changes in the economy caused by the pandemic. Being aware of this, we want to continue our research in the future. To articulate this in the paper we have added a paragraph on the limitations of our research (at the time we conducted it) and information on our future intentions (line 587 to 591).

Sincerely,

Magdalena Kozera-Kowalska (corresponding author)

Jarosław Uglis, Jarosław Lira

---

## [Editor Report · Decision Letter 1]

14 May 2021

A framework to measure the taxonomic of economic anchor: a case study of the Three Seas Initiative countries

PONE-D-21-02343R1

Dear Dr. Magdalena Kozera-Kowalska,

We’re pleased to inform you that your manuscript has been judged scientifically suitable for publication and will be formally accepted for publication once it meets all outstanding technical requirements.

Kind regards,

Carlos Alberto Zúniga-González, Ph.D

Academic Editor

PLOS ONE

Additional Editor Comments (optional):

We thank the authors for the improvements made and the effort to improve the quality of their manuscript.
---

## [Editor Report · Acceptance letter]

19 May 2021

PONE-D-21-02343R1 

A framework to measure the taxonomic of economic anchor: a case study of the Three Seas Initiative countries 

Dear Dr. Kozera-Kowalska:

I'm pleased to inform you that your manuscript has been deemed suitable for publication in PLOS ONE. Congratulations! Your manuscript is now with our production department. 

Kind regards, 

on behalf of

Dr. Prof. Carlos Alberto Zúniga-González 

Academic Editor

PLOS ONE